# Oral Health-Related Quality of Life Changes after Clinical Remounting of Existing Dentures

**DOI:** 10.3390/healthcare10101960

**Published:** 2022-10-07

**Authors:** Chi-Hsiang Cheng, Ikiru Atsuta, Kiyoshi Koyano, Yasunori Ayukawa

**Affiliations:** 1Section of Implant and Rehabilitative Dentistry, Division of Oral Rehabilitation, Faculty of Dental Science, Kyushu University, Fukuoka 8128582, Japan; 2Division of Advanced Dental Devices and Therapeutics, Faculty of Dental Science, Kyushu University, Fukuoka 8128582, Japan

**Keywords:** clinical remount, complete denture, malocclusion, OHIP-EDENT

## Abstract

The clinical remount is an accurate and efficient way to reset the occlusion of delivered removable dentures if major occlusal correction is required. Although previous studies have reported that clinical remounting of existing dentures enhances patients’ oral function, little subjective feedback is available. This retrospective study reports short-term changes in oral-health-related quality of life (OHRQoL) and masticatory function after clinical remounting of existing dentures. Three time points were defined: before adjustment (T0), immediately after adjustment (T1), and 1 week after adjustment (T2). The medical records of seven patients were analyzed. The mean age of participants was 77.71 years, and the mean service period of their prostheses was 9.43 months. The mean scores of the OHIP-EDENT-J questionnaire at the respective time points were 35, 21.14, and 22.14. The mean readings of masticatory function at the respective time points were 76.71, 89.29, and 111.86. Significant differences in the OHIP-EDENT-J were found between T0 and T1, and T0 and T2; and in masticatory function between T1 and T2, and T0 and T2. The results indicated that after rebalancing of the occlusion of the existing dentures, the patient-reported OHRQoL was improved immediately and maintained at least for a short time, and masticatory function was enhanced over a 1-week period.

## 1. Introduction

In complete denture treatment, an accurate, reproducible jaw relationship plays an important role in achieving patient satisfaction [1]. However, previous studies have reported that even when the occlusion and articulation of the dentures are carefully balanced during the delivery process, the occurrence of malocclusion seems inevitable after a period of service [2,3,4,5]. If rebalancing of the occlusion is necessary, clinicians can achieve the goal with either intraoral or extraoral methods. Although minor occlusal flaws can be corrected with a direct intraoral method, the resilience of the denture-supporting tissues, displacement of the denture base, and saliva may hamper the adjustment if major corrections are needed [6].

Previous review examined the extraoral method, also known as the clinical remount procedure, an accurate and efficient way to correct cumulative errors in the jaw relationship [7]. Schierano et al. reported that a patient’s thickness discrimination ability was enhanced after a clinical remount was performed on existing dentures [5]. Kawahara et al. used standardized gummies to evaluate masticatory function before and immediately after a clinical remount, and concluded that the masticatory function was significantly higher after the occlusal adjustment [8]. Although previous research has reported that performing clinical remount procedures on existing dentures enhances oral function, subjective feedback from patients is still scarce.

Various types of questionnaires can be employed to investigate patient feedback about received treatments [9,10,11]. The Oral Health Impact Profile (OHIP) was originally designed to measure patients’ perceptions of oral-health-related quality of life (OHRQoL) [12,13]. Locker and Allen further adapted the original OHIP into the Oral Health Impact Profile for Edentulous Patients (OHIP-EDENT) [14]. The OHIP-EDENT is considered the gold standard for reporting patient-centered quality of life in edentulous patients. It contains 19 questions describing the impacts of oral-health-related problems on daily activities. Participants respond by rating the frequencies of the described problems (4 = always, 0 = never). The total score of this questionnaire can be from 0 to 76. A higher score indicates a lower OHRQoL. This questionnaire has been presented as a primary outcome in multiple randomized controlled trials (RCTs) concerning complete dentures [15].

This study aimed to observe short-term changes in the OHRQoL as measured by the OHIP-EDENT questionnaire after rebalancing the occlusion of existing dentures. The null hypothesis was that the OHRQoL score would not change significantly after the clinical remounting procedure.

## 2. Materials and Methods

### 2.1. Study Population

This retrospective, observational clinical pilot study was conducted in the Department of removable prosthodontics, Kyushu University Hospital, Japan, with no interventions. The necessity for the clinical remount procedure was evaluated by first-line clinicians. Approval from the Institution Review Board of Kyushu University Hospital (21169-00) was obtained before data extraction. After data extraction, all enrolled participants were informed about the project at their regular follow-up appointments. Informed consent was acquired from all participants.

### 2.2. The Clinical Remount Procedure

The clinical remount procedure performed at Kyushu University Hospital has been described in our previous study [16]. The tissue surfaces under the dentures were checked and adjusted if indicated before bite registration was taken. Bimaxillary dentures were then stabilized, and centric relation (CR) was guided and recorded with bite wax (Bitewax; GC, Tokyo, Japan). The bite registration was then carefully inspected and verified intra-orally again. If any perforation was noted or the bite registration could not be verified, a new bite registration was taken [17].

The dentures were then arbitrarily mounted on a type-3, non-arcon mean-value articulator with a mounting platform. The condylar guidance and Bennett angle were set to the average value and kept unchanged during the whole procedure [16]. The incisal pin was then removed, and premature contacts were marked with 25 μm-thick articulating paper (Precut Articulating Paper; Morita, Osaka, Japan). Occlusal contact points of the dentures were defined by tactile sensation. If the articulating paper could be pulled out from the occluded dentures without tearing, the contact was considered to be loose, and premature contacts were further removed [18]. Selective grinding was then carried out to establish bilateral balanced occlusion [19].

After the clinical remount procedure, follow-up appointments were arranged in accordance with the preferences and schedules of the first-line clinicians and patients.

### 2.3. Data Collection

Two main data sources, masticatory function and the questionnaire, were obtained and analyzed from the medical record system of Kyushu University Hospital by a single researcher (C.H.C.).

Masticatory function and occlusal force were evaluated in accordance with the instructions of The Japanese Universal Health Insurance Coverage System (JUHICS) [20]. The patient was instructed to chew 2 g of gummy jelly freely for 20 s, and then rinse with 10 mL water. The fragmented gummy jelly and water were then spat out, and the amount of dissolved glucose was measured in mg/dL (Gluco Sensor GS-II; GC, Tokyo, Japan) [21].

The questionnaire used in the current study was the Japanese version of the OHIP-EDENT (OHIP-EDENT-J) [22]. The retrieved OHIP-EDENT-J data were further analyzed for three major factors: physical impact, psychological impact, and social impact, as described by Possebon et al. [23]. The physical impact subscales contained 10 questions, making a total possible score of 40. The total scores for psychological impact and social impact were 16 and 20, respectively.

### 2.4. Data Extraction Method and Statistics Analysis

By studying the distribution of follow-up appointments, three time points were established: pre-treatment (T0), immediately after the clinical remount (T1), and 1 week after the clinical remount (T2). Between April 2019 and July 2022, 97 clinical remount procedures were carried out. Fifty cases in which the clinical remount procedure was performed at the appointment of delivery of the new prosthesis were excluded. Another 40 cases were excluded because the data for the OHIP-EDENT-J and masticatory function at T0, T1, and T2 were not fully recorded. Eventually, 7 individual cases were analyzed in the current study (Figure 1). General information about the participants, including sex, age, and condition and service period of the current prostheses, was recorded.

Paired data for masticatory function and the OHIP-EDENT-J and its subscales were analyzed with a one-tailed Wilcoxon sign-rank test between time points (e.g., T0T1, T0T2, T1T2). A probability value of less than 0.05 was considered statistically significant.

## 3. Results

### 3.1. General Information

The medical records of two men and five women, aged 69 to 87 years (mean: 77.71 years, median: 79 years), were analyzed. Case numbers were allocated, 01 to 07, and their characteristics are listed in Table 1. Two participants had bimaxillary total edentulism, and five had partial edentulism. The service period of the prostheses was 6.5 to 17 months (mean: 9.43 months, median: 7.5 months). The medical records indicated that cases 03, 06, and 07 had undergone a clinical remount adjustment for the current prostheses at the delivery appointment.

### 3.2. Masticatory Function

All acquired data are documented in Table 2. The changes in masticatory function between time points were analyzed (Figure 2). Compared with T0, the mean value of T1 increased by 16.4%, and the increments between T1 and T2 and between T0 and T2 were 25.28% and 45.82%. Although there was no statistically significant difference in masticatory function between before and immediately after the clinical remount procedure, significant differences were noted between T1 and T2 and between T0 and T2.

### 3.3. OHIP-EDENT-J Questionnaire

Changes in the OHIP-EDENT-J between time points were analyzed (Figure 3). Compared with T0, the mean value score of the OHIP-EDENT at T1 decreased by 13.86 points and slightly increased by 1 point at T2. Significant differences were noted between T0 and T1 and between T0 and T2, but not between T1 and T2.

The seven subscales of the OHIP questionnaire were then further divided into three main factors for further analysis—physical, psychological, and social (Table 3 and Figure 4, Figure 5 and Figure 6). Compared with T0, the total scores for the physical and psychological impact at T1 dropped by 11 and 0.86 respectively, whereas the total score for social impact increased by 1.42. A significant difference was only found for physical impact among the three major factors.

Compared with T1, the scores for the physical and psychological impact at T2 increased by 2.43 and 0.28, respectively, whereas the total score for social impact decreased by 1.85. There were no significant differences in any of the three factors between T1 and T2, and physical impact was the only factor that had a significant difference between T0 and T2.

Our analysis led us to reject the null hypothesis of the current study, indicating that after a clinical remount is performed on existing dentures, the OHRQoL will improve and maintain this improvement for at least 1 week.

## 4. Discussion

A harmonious occlusion and an accurate, reproducible jaw relationship are important factors for dentures to function efficiently [6]. However, successful rehabilitation often relies more on positive relationships between the clinician and the patient [24]. Therefore, along with objective clinical findings, it is essential to collect patients’ subjective feedback for a comprehensive evaluation.

This study aimed to reveal the merits of performing clinical remount procedures on existing dentures by assessing patients’ OHRQoL. According to the previous review, rebalancing of the occlusion was indicated only if patients reported occlusion-related complaints with their existing dentures [7]. Therefore, it would be extremely difficult and unethical to intentionally create such niche treatment needs to conduct well-controlled, prospective research. The retrospective, observational design of the current study provided a cost-effective way to reveal some preliminary data for subsequent research; however, the evaluation tools were restricted by the available data.

Data from the masticatory function test were also extracted to ensure that the clinical remount procedures were properly executed. In contrast with Kawahara et al., masticatory function immediately after occlusal adjustment failed to show significant improvement in the current study [8]. Nevertheless, considering that the prosthesis designs varied, participants were treated by different clinicians with varying treatment plans, and positive improvements were recorded between T0 and T2 and between T1 and T2, the authors believe that all participants received valid treatment. Additionally, previous research reported by Kawahara et al. only compared masticatory function before and immediately after the clinical remounting [8]. The current study indicated that masticatory function changed incrementally during the one-week follow-up period.

To our knowledge, the current study was the first to report patients’ subjective perceptions after their existing dentures were rebalanced. The mean pre-operative baseline OHIP-EDENT scores were reported to be 28.63 with 95% confidence intervals from 21.93 to 35.34 [15]. In the current study, the mean OHIP scores before treatment were higher than those at baseline, indicating the necessity for maintenance of the participants’ prostheses.

Further analysis showed that the physical impact subscale of the OHRQoL was the only one that showed a significant reduction after the clinical remount procedure. As textbooks and previous studies have documented, faulty occlusion may cause tissue irritation, unstable dentures, and other complaints [6,17,25]. Therefore, it was understandable that the physical hindrance was reduced after the clinical remount procedure. Additionally, there were only minimal changes in the occlusal surface of the rebalanced dentures; and no significant changes in overall esthetics, artificial teeth arrangement, and flange extension; it was unsurprising that there were no significant changes in psychological or social impacts in such a short follow-up period.

The main limitations of the current study were the small study population and the short follow-up time. Previous studies have suggested that occlusal force and masticatory performance keep improving with the continuous use of newly inserted dentures over the first few months [26,27]. Further research with more participants and a longer follow-up period for both OHRQoL and masticatory function is necessary to determine the treatment’s efficiency and a valid timeframe for rebalancing the occlusion of existing dentures.

The need for complete denture treatment is not likely to decrease in the near future [28]. The authors consider the clinical remount technique to be a useful tool for fabricating quality dentures and maintaining those already in use.

## 5. Conclusions

This pilot study revealed that performing a clinical remount procedure on patients’ existing dentures reduced the scores on the OHIP-EDENT-J questionnaire immediately after the adjustment and enhanced masticatory function within the 1-week follow-up period. Our findings indicated that after the occlusion of the existing dentures was rebalanced, the patient-reported OHRQoL was potentially enhanced and maintained for a short period of time.

## Figures and Tables

**Figure 1 healthcare-10-01960-f001:**
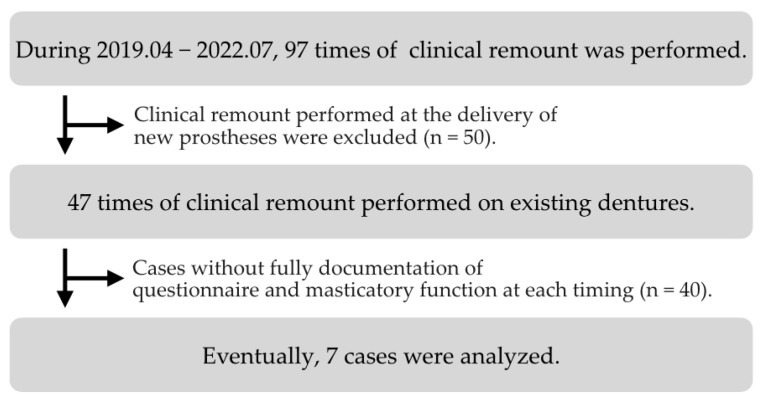
Screening process for data extraction.

**Figure 2 healthcare-10-01960-f002:**
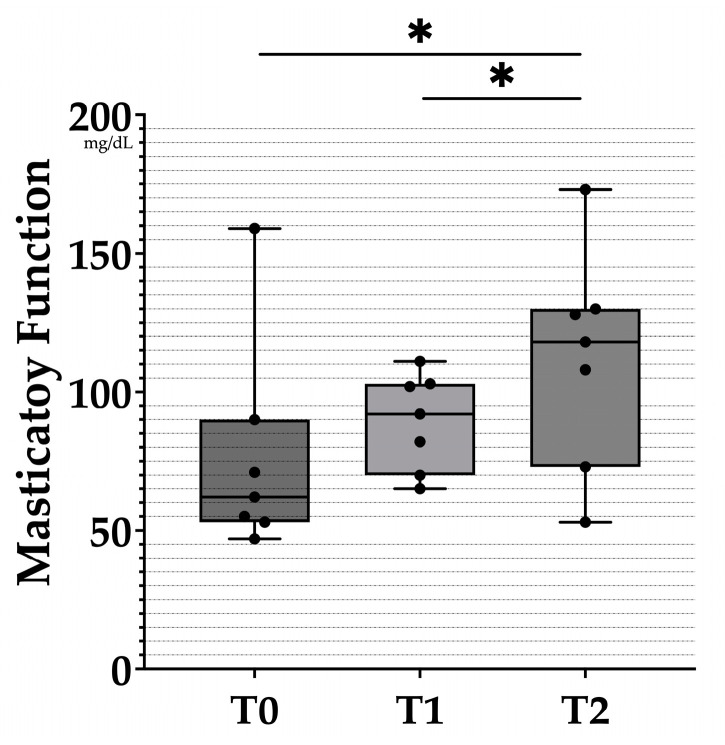
Box plots of masticatory function at each time point. The black dots represent the distribution of raw data. T0: before the clinical remount. T1: immediately after the clinical remount. T2: one week after the clinical remount. Statistical analysis was performed with the Wilcoxon sign-rank test. Predetermined level of significance: *p* < 0.05 (*).

**Figure 3 healthcare-10-01960-f003:**
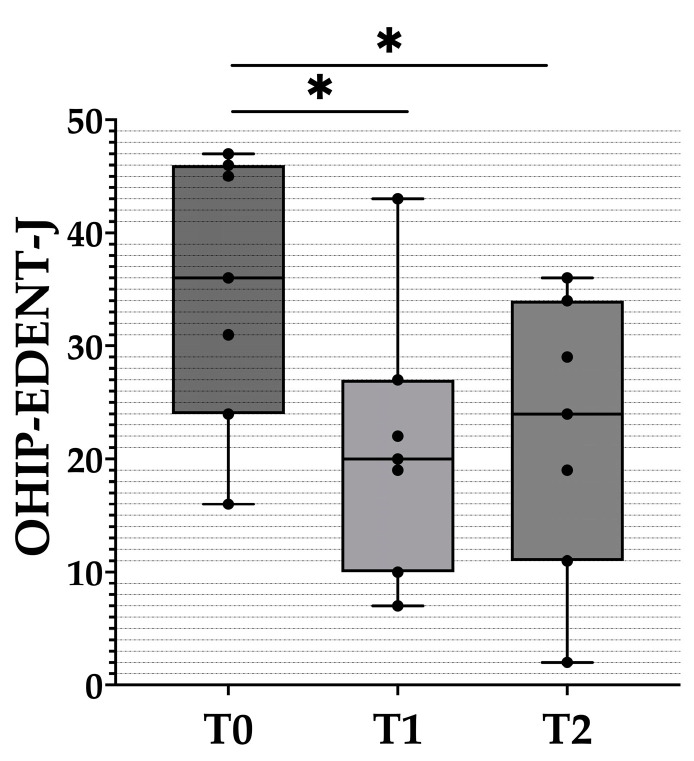
Box plots figure of the OHIP-EDENT-J at each time point. The black dots represent the distribution of raw data. T0: before the clinical remount. T1: immediately after the clinical remount. T2: 1 week after the clinical remount. Statistical analysis was performed with the Wilcoxon sign-rank test. Predetermined level of significance: *p* < 0.05 (*).

**Figure 4 healthcare-10-01960-f004:**
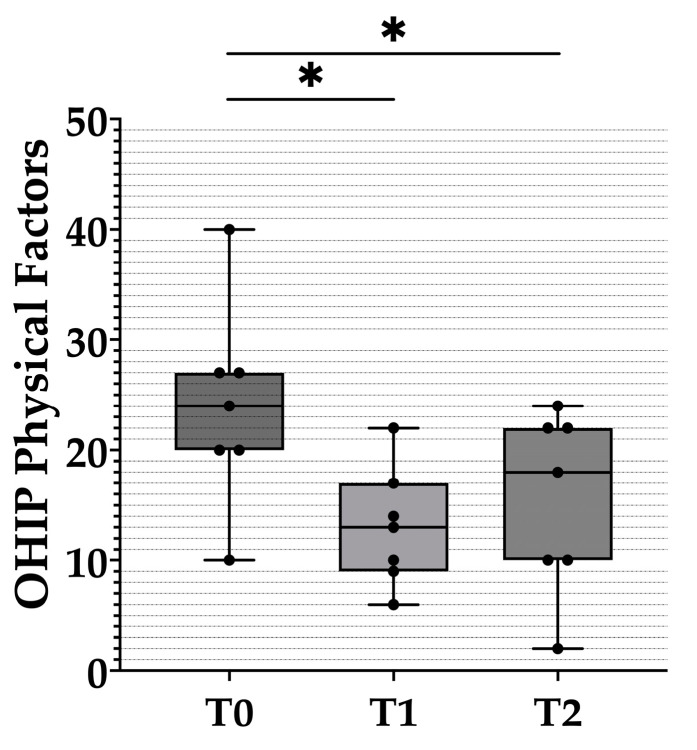
Box plots of the physical impact scores from the OHIP-EDENT-J questionnaire. The black dots represent the distribution of raw data. T0: before the clinical remount. T1: immediately after the clinical remount. T2: 1 week after the clinical remount. Statistical analysis was performed with the Wilcoxon sign-rank test. Predetermined level of significance: *p* < 0.05 (*).

**Figure 5 healthcare-10-01960-f005:**
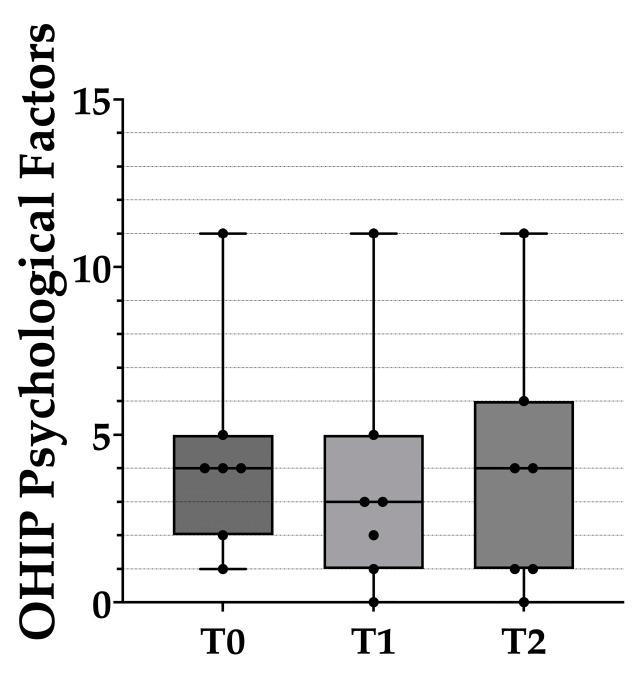
Box plots of the psychological impact scores from the OHIP-EDENT-J questionnaire. The black dots represent the distribution of raw data. T0: before the clinical remount. T1: immediately after the clinical remount. T2: 1 week after the clinical remount. Statistical analysis was performed with the Wilcoxon sign-rank test. Predetermined level of significance: *p* < 0.05.

**Figure 6 healthcare-10-01960-f006:**
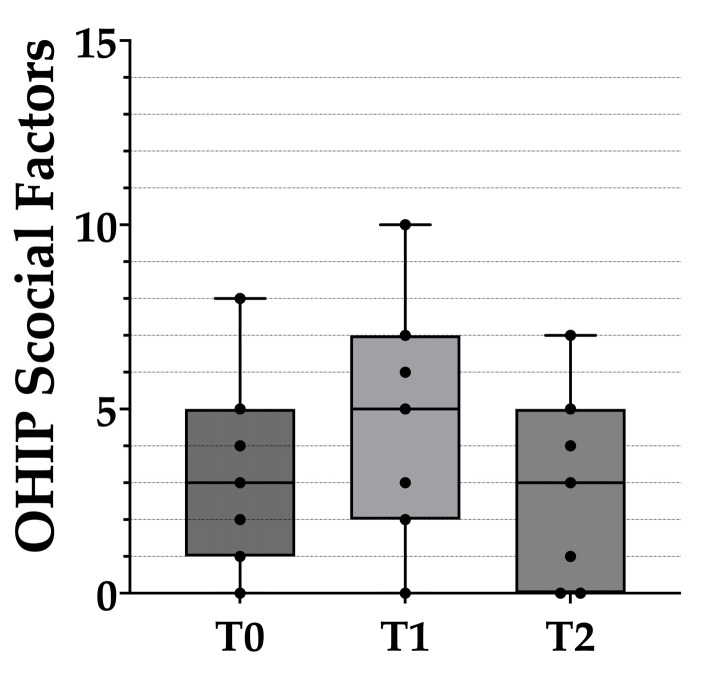
Box plots of the social impact scores from the OHIP-EDENT-J questionnaire. The black dots represent the distribution of raw data. T0: before the clinical remount. T1: immediately after the clinical remount. T2: 1 week after the clinical remount. Statistical analysis was performed with the Wilcoxon sign-rank test. Predetermined level of significance: *p* < 0.05.

**Table 1 healthcare-10-01960-t001:** General characteristics of analyzed cases. CD: complete denture. RPD: removable partial denture.

Case	Gender	Age	Maxillary Prosthesis	Mandibular Prosthesis	Clinical Remounted at the Delivery Appointment	Duration Since Rehabilitation (Months)
01	F	81	CD	CD	No	7.5
02	M	69	CD	CD	No	17
03	F	74	CD	RPD	Yes	12
04	F	79	RPD	RPD	No	6.5
05	F	82	CD	RPD	No	8.5
06	M	72	RPD	RPD	Yes	7.5
07	F	87	CD	RPD	Yes	7
Mean (SD)		77.71 (2.39)				9.43 (1.44)

**Table 2 healthcare-10-01960-t002:** All acquired data, mean values, and standard deviations. T0: before the clinical remount. T1: immediately after the clinical remount. T2: 1 week after the clinical remount. MF: masticatory function measured with the Gluco Sensor GS-II in mg/dL. OHIP: OHIP-EDENT-J questionnaire.

Case	T0		T1		T2	
	M.F.	OHIP	M.F.	OHIP	M.F.	OHIP
01	62	36	92	22	118	29
02	71	45	103	19	128	24
03	47	46	70	43	53	36
04	55	16	82	20	108	19
05	53	31	65	27	73	34
06	159	47	102	7	173	11
07	90	24	111	10	130	2
Mean (SD)	76.71 ± 14.73	35 ± 4.54	89.29 ± 6.63	21.14 ± 4.48	111.86 ± 14.92	22.14 ± 4.69

**Table 3 healthcare-10-01960-t003:** All data acquired from the OHIP-EDENT-J questionnaire were further divided into three major factors. T0: before the clinical remount. T1: immediately after the clinical remount. T2: 1 week after the clinical remount. Phy (40): Physical impact. The total score of this subscale was 40. Psy (16): Psychological impact. The total score of this subscale was 16. SC (20): Social impact. The total score of this subscale was 20.

Case	T0			T1			T2		
	PHY (40)	PSY (16)	SC (20)	PHY (40)	PSY (16)	SC (20)	PHY (40)	PSY (16)	SC (20)
01	27	5	4	13	3	6	18	4	7
02	40	2	3	14	2	3	22	1	1
03	27	11	8	22	11	10	22	11	3
04	10	1	5	10	5	5	10	4	5
05	24	4	2	17	3	7	24	6	4
06	20	4	1	6	1	0	10	1	0
07	20	4	0	9	0	2	2	0	0
Mean (SD)	24 (3.46)	4.43 (1.21)	3.29 (1.02)	13 (2.02)	3.57 (1.38)	4.71 (1.27)	15.43 (3.11)	3.85 (1.44)	2.86 (1.01)

## Data Availability

The data presented in this study are available on request from the corresponding author. The data are not publicly available due to the privacy of patients. All data was stored and managed under the instructions of the Institutional Review Board of Kyushu University Hospital.

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
