# Peer review of "Oral Health-Related Quality of Life Changes after Clinical Remounting of Existing Dentures"

_healthcare, 2022, doi:10.3390/healthcare10101960_

Round 1

Reviewer 1 Report

This study titled “OHRQoL changes after clinical remounting of existing dentures” aimed to analyze the patients’ perception of the quality of life after having the denture remounted.
The study is interesting and well-written, but it was raised some questions (below).

line 39. “(p. 395)” - remove it
line 80. “(pp. 269)” - remove it
line 214 - similar

- Why was this study considered a pilot and retrospective study?

- Did the authors make the sample size calculation? I know that it is a pilot study; however, 7 cases I considered insufficient to do statistical analysis and a conclusion like presented.

- Who were the professionals responsible to collect data? Include this info in the text.

Author Response

The modification report for Reviewer 1

Dear Reviewer 1:

Thank you for pointing out the flaws of our article.
    We had modified the manuscript according to your comment.

  1. The citations in lines 39, 80, and 214 were modified.
    The researcher corresponded data collection was attached in the text. (Chapter 2.3)
    Thank you for making our article clear.

  1. Thank you for your kind question. We agreed with your considerations about the sample size. According to our data extraction method, only 7 fully documented cases were able to be analyzed in the past 3 years. Therefore, there was only a minimum allowance for performing the sample size calculation in the current study.

    As we mentioned in our article, this article is the first study that reported the subjective feedback of patients after their existing dentures were re-balanced. Because of scarce related research and uncertain results, we considered retrospective design would fit this pilot study with a small population the most.

Last but not least, the current study revealed the mean scores and standard deviations of the OHIP-EDENT questionnaire at different time points in the statistical population of Kyushu University Hospital, which were considered useful parameters for planning further research projects. We sincerely hope these preliminary results bring more attention and resources to this fascinating research field.

Once again, we appreciate your patient reading and delicate check for our manuscript.

Sincerely,

Chi-Hsiang Cheng & Ikiru Atsuta

Reviewer 2 Report

article: OHRQoL changes after clinical remounting of existing 2
dentures

Many thanks for paper submission to healthcare journal. I appreciate the authors work, however some small modifications are needed for acceptance.

1) Please modify the title with full name of your OHRQL parameters: please change to " oral health-related quality of life changes after clinical remounting of existing dentures": this helps the reader to link the paper to existing literature of OHRQL.
2) at line 52 please add the following references regards to the use of OHRQL in third molar surgery

Chisci G, De Felice C, Parrini S, Signorini C, Leoncini S, Ciccoli L, Volpi N, Capuano A. The role of preoperative oxidative stress and mandibular third molar postoperative outcome. Int J Oral Maxillofac Surg. 2013 Nov;42(11):1499-500. doi: 10.1016/j.ijom.2013.07.003. Epub 2013 Aug 9. PMID: 23932577.

Parrini S, Chisci G, Leoncini S, Signorini C, Volpi N, Capuano A, Ciccoli L, De Felice C. F2-Isoprostanes in soft oral tissues and degree of oral disability after mandibular third molar surgery. Oral Surg Oral Med Oral Pathol Oral Radiol. 2012 Sep;114(3):344-9. doi: 10.1016/j.oooo.2012.03.010. PMID: 22862975.

3) reference 28 is not present in the bibliography, please correct

4) at line 261 please remove the phrase "Even with only a limited study population and a short follow-up time"

Author Response

The modification report for Reviewer 2

Dear Reviewer 2:

Thank you for pointing out the flaws of our article. We had modified the manuscript according to your comment.

  1. The title of our article had been changed into
    “Oral health-related quality of life after clinical remount of existing dentures.”
    Thank you for making our article clear to readers.

  1. Thank you for sharing such interesting research with us.

The following article had been cited in our article:

Parrini S, Chisci G, Leoncini S, Signorini C, Volpi N, Capuano A, Ciccoli L, De Felice C. F2-Isoprostanes in soft oral tissues and degree of oral disability after mandibular third molar surgery. Oral Surg Oral Med Oral Pathol Oral Radiol. 2012 Sep;114(3):344-9.

However, we found that the other provided reference has little relation to OHQoL. Hence, careful evaluations should be considered before citing the article titled “The role of preoperative oxidative stress and mandibular third molar postoperative outcome.”

We are glad to hear from you and cite this article if requested.

  1. The bibliography of the article had been checked and modified.

  1. The conclusion of the article had been modified according to your suggestion.
    Thank you for making our work more concise!

Once again, we appreciate your patient reading and delicate check for our manuscript.

Sincerely,

Chi-Hsiang Cheng & Ikiru Atsuta